# Architectural Sweet Spots for Modeling Human Label Variation by the Example of Argument Quality: It's Best to Relate Perspectives!

**Philipp Heinisch**[1*], **Matthias Orlikowski**[1*], **Julia Romberg**[2], and **Philipp Cimiano**[1]

[1]Center for Cognitive Interaction Technology (CITEC), Bielefeld University, Germany
{pheinisch,morlikowski,cimiano}@techfak.uni-bielefeld.de
[2]Department of Social Sciences, Heinrich Heine University Düsseldorf, Germany
julia.romberg@hhu.de

## Abstract

Many annotation tasks in natural language processing are highly subjective in that there can be different valid and justified perspectives on what is a proper label for a given example. This also applies to the judgment of argument quality, where the assignment of a single ground truth is often questionable. At the same time, there are generally accepted concepts behind argumentation that form a common ground. To best represent the interplay of individual and shared perspectives, we consider a continuum of approaches ranging from models that fully aggregate perspectives into a majority label to "share nothing"-architectures in which each annotator is considered in isolation from all other annotators. In between these extremes, inspired by models used in the field of recommender systems, we investigate the extent to which architectures that include layers to model the relations between different annotators are beneficial for predicting single-annotator labels. By means of two tasks of argument quality classification (argument concreteness and validity/novelty of conclusions), we show that recommender architectures increase the averaged annotator-individual $F_1$-scores up to $43\%$ over a majority-label model. Our findings indicate that approaches to subjectivity can benefit from relating individual perspectives.

## 1 Introduction

There is inherent subjectivity in many annotation tasks in natural language processing (Shahid et al., 2020; Kumar et al., 2021; Thorn Jakobsen et al., 2022). Recent work has criticized the widely adopted approach of viewing variation in human labeling behavior as "noise" (Plank, 2022), advocating against approaches that disregard the richness of human annotations and perspectives by aggregating them into a single label. In fact, it has been argued that disagreement should not be regarded

---

*Authors contributed equally to this work.

as a problem, but rather as a chance to end up with more user-adaptable classifiers, giving voice to minorities as well (Prabhakaran et al., 2021; Gordon et al., 2022).

In this paper, we hypothesize that machine learning can best address variation in human labeling by both accounting for subjective perspectives of individuals and for more objective concepts that build a common ground between annotators. A prime example of the interplay of individual and shared perspectives is the understanding and assessment of argumentation and, in particular, of argument quality (Romberg, 2022). Given the subjectivity that many concepts of argument quality face, it has been generally shown that its annotation often results in only fair to moderate inter-annotator agreements (Aharoni et al., 2014; Rinott et al., 2015; Habernal and Gurevych, 2017; Shnarch et al., 2018).

For example, Gretz et al. (2020) determined a global value of argument quality by asking annotators "if they would recommend a friend to use that argument as is in a speech supporting/contesting the topic". Even in more well-defined aspects of argument quality, such as the sufficiency of an argumentative conclusion, Stab and Gurevych (2017) observed "hard cases" in which labeling depends on subjective interpretations of keywords without being able to agree on a single ground truth. However, alongside "hard cases" of irreconcilable individual perspectives, in assessing argument quality we regularly also observe uncontroversial cases such as in human annotation of the validity and novelty of conclusions (Heinisch et al., 2022) or the concreteness of arguments (Romberg et al., 2022).

In order to verify our hypothesis, we investigate the impact of different architectural design choices on the ability of models to predict the labels of single annotators. To this end, we look at a continuum of model architectures between such that fully suppress annotation variation by learning

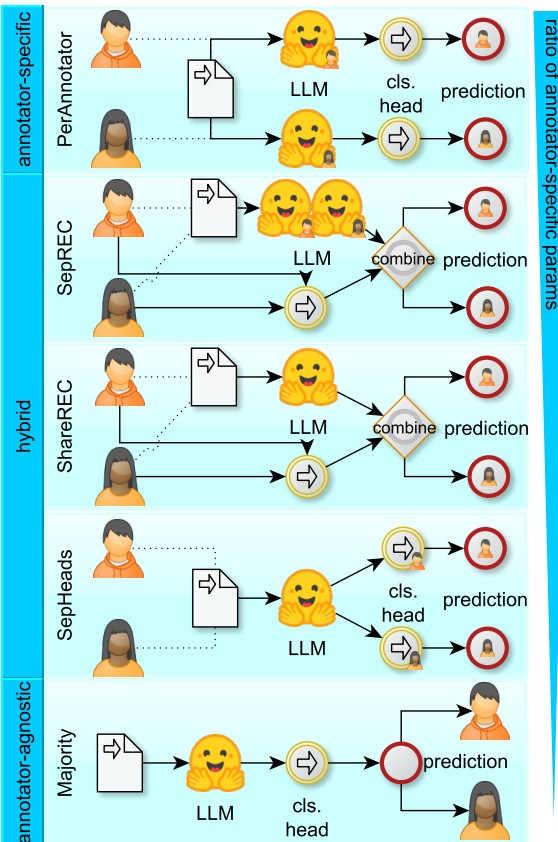

Figure 1: Overview of our different approaches modeling annotator-(a)gnostic behaviors.

from an aggregated label and such that are specifically tailored to each annotator. Along this continuum, we focus in particular on models that include components that involve the perspectives of single annotators (Davani et al., 2022), or are able to find and relate similar annotation behavior shared among different annotators using a recommender approach (Gordon et al., 2022). Figure 1 gives a first overview of the continuum of concepts and models, which we explain in detail in Section 3.

Drawing on two tasks of argument quality, namely argument concreteness and validity/novelty of conclusions, we show that architectures inspired by recommender systems perform best in learning from disagreement. We attribute this to the identification of patterns in annotation behavior across individual annotators while being faithful to single choices.

Our main contributions are:

- We present a novel framework[1] for comparing different architectural design choices in or-

der to model individual label decisions along the continuum from predicting a single label for all annotators to "share nothing" architectures that model the label decisions of single individuals independently from each another. *This framework is not limited to our use case of argument quality, but can be applied to any classification task whose annotation combines subjective and objective aspects, in order to find architectural sweet spots for modeling label variation.*

- We perform an extensive automatic evaluation tuning different model types and hyperparameters on two datasets corresponding to three tasks, involving vanilla large language models (LLM) as well as LLMs with annotator-specific classification heads and recommender models adapted to the classification tasks in argument quality. Using a recommender-based architecture, we increase the averaged annotator-individual $F_1$-scores by up to 4 points in the classification of argument concreteness, 18 points in the classification of conclusion validity and 19 points in the classification of conclusion novelty.

- We conduct a qualitative case study of the behavior of our models on controversial examples in order to shed light on the differences regarding the effect of annotator-(dis)agreement, differing amounts of annotated samples between annotators, and annotation behavior.

Extending previous work that proposed to take subjectivity into account by predicting the degree of expected label variation by Romberg (2022), our study presents the first approach in the field of argument mining to learn directly from the individual human labels. Beyond the specific contributions mentioned above, our work might encourage future research in argument mining and other fields to step away from systems that suppress valid perspectives by relying on a single aggregated label, and instead moving on towards systems that cover the multi-faceted spectrum of opinions.

## 2 Related Work

### 2.1 Subjectivity & Modeling Individual Annotators

There is a growing body of work researching subjectivity (Ovesdotter Alm, 2011; Rottger et al.,

---

[1] https://github.com/phhei/RelatePerspectives-sweetspots

2022), learning with disagreement (Uma et al., 2021a; Leonardelli et al., 2023; Sandri et al., 2023), diversity of perspectives (Abercrombie et al., 2022; Cabitza et al., 2023) and human label variation (Plank, 2022). Despite the varying terminology, these works overlap in concerns that aggregating labels into a single "truth" is not appropriate for many tasks (Aroyo and Welty, 2015; Uma et al., 2021b; Basile et al., 2021) and might not represent perspectives equally (Prabhakaran et al., 2021; Abercrombie et al., 2022). This line of research has produced various approaches to learning models based on individual annotations (Plank et al., 2014; Jamison and Gurevych, 2015; Akhtar et al., 2020; Fornaciari et al., 2021; Cercas Curry et al., 2021; Plepi et al., 2022).

A particular way of learning from annotator-specific labels are models which learn to predict individual annotators' decisions. Conceptually, these models can be seen as feature-based models of annotation (see for an overview Paun et al. 2022b) in that they model how each annotator labels individual examples. However, in contrast to standard models of annotation (e.g., Hovy et al. 2013; Passonneau and Carpenter 2014; Paun et al. 2018), their goal is not to aggregate decisions to a single label before training but to learn classifiers directly from non-aggregated annotations (Paun et al., 2022a). Most work in this area (Raykar et al., 2010; Albarqouni et al., 2016; Guan et al., 2018; Rodrigues and Pereira, 2018) argues for training on non-aggregated labels as a way to deal with varying annotator reliability to better derive the correct labels. Among these, Chu et al. (2021) explore an idea similar to our emphasis of common ground in addition to individual variation, modeling annotation noise in terms of both individual and common noise. Closely related to our work, more recent studies (Davani et al., 2022; Gordon et al., 2022) focus on subjective tasks for which a single ground truth can not always be determined.

Davani et al. (2022) introduce a multi-annotator model (further explored in Orlikowski et al. 2023; Vitsakis et al. 2023), in particular a variant using a multi-task architecture: For each annotator, there is a separate classification head trained on annotations from that annotator. All these annotator layers share a pre-trained language model used to encode the input. We evaluate this architecture in our experiments. Gordon et al. (2022) present a model that also predicts individual annotations and

allows a user to interactively aggregate them based on "a jury" inspired by the US judicial system. Their approach is based on a recommender architecture using "Deep & Cross Networks" (Wang et al., 2021) which we also use in our work.

## 2.2 Subjectivity in Argument Mining

Argumentation and, in particular, the human understanding of its quality, is often subjective, conditioned by a variety of phenomena (e.g., van der Weide et al., 2010; Esau, 2018). Only recently has the argument mining community joined other disciplines, such as social sciences and formal argumentation, in addressing the backing mechanisms for differing perspectives in argumentation.

Ajjour et al. (2019) first examined the framing of arguments in order to appeal to specific audiences based on their interests, cultural backgrounds, and socialization. Putting a special focus on moral frames, Kobbe et al. (2020) and Alshomary et al. (2022) studied different moral belief systems that arguments are subject to, drawing on the moral foundations theory (Haidt and Joseph, 2004).

Going into more detail on individuals' diverse beliefs and emphasises, Kiesel et al. (2022) utilized computational methods to identify a comprehensive taxonomy of $54$ human values (Searle, 2003) embedded in arguments. In addition, the effect of storytelling on individual perceptions of argumentation was addressed by Falk and Lapesa (2022).

Besides the motives behind subjective reasoning, research also considered systems that include multiple perspectives in the output, such as a diversity of stances about some claim (Chen et al., 2019).

A direct modeling of different perspectives as part of the machine learning process itself, on the other hand, has hardly been considered so far. The only contribution in this direction was made by Romberg (2022), who presented a methodology for integrating subjectivity information into conventional text classification workflows of argument mining. While that approach involves training a separate classifier to predict a subjectivity value, in this work we implement models that directly learn from the non-aggregated labels.

## 3 Methodology

We aim at the comparison of different paradigms that can be applied in order to model a classifier for predicting individual labeling decisions in subjective annotation tasks. To this end, we define a

model spectrum ranging from a purely annotator-specific model design (denying that there is anything learnable which is shared among the annotators, i.e., the objective grounding) to a purely annotator-agnostic model design which only considers the majority vote (denying that there is anything learnable which is individual for an annotator, i.e., the subjective grounding).

We also include models that exploit the characteristics of both sides of the coin. These hybrid models combine model components that are shared for all annotators as well as model components that are specific for each annotator. In this way we can explore different architectural design choices along the above mentioned continuum, varying the degree to which the shared labeling behavior of certain groups of users is modeled in the architecture.

The proposed model spectrum complements existing taxonomies of learning from disagreement. Uma et al. (2021b) distinguish four categories of approaches, one of them being "learning directly from crowd annotations". Within this category, our proposed spectrum allows to further differentiate how models process individual annotations.

Figure 1 shows our broad bandwidth of approaches, which covers five different approaches to classification: a majority vote model, per-annotator models, and three hybrid approaches, namely a model with annotator-specific classification heads, a recommender system with a shared text encoder, and a recommender system with annotator-separated text encoders. These will be described in more detail in the following subsections.

### 3.1 Two poles: annotator-specific and annotator-agnostic approaches

For the pure annotator-specific and annotator-agnostic approaches, we consider a pre-trained LLM with a standard classification head. The only difference between the two paradigms is the way the dataset is preprocessed. In the case of the "share nothing" annotator-specific paradigm, the dataset is split annotator-wise, i.e., one split consists of all annotated instances paired with the individual annotation of exactly one annotator. Hence, having $n$ different annotators, we fine-tune $n$ language models, assuming training data for each annotator. We call this variant *PerAnnotator*. The opposite annotator-agnostic paradigm considers the full dataset with majority-aggregated annotations, resulting in a single annotator-agnostic model that

captures the majority perspective (*Majority*).

### 3.2 Approaches *between* annotator-specific and annotator-agnostic approaches

For modeling both objective and subjective components, we compare two different architectures.

**Annotator-specific classification head** The first architecture replaces the single classification head of the LLM with a set of annotator-specific classification heads. This architecture is equivalent to (multi-task) multi-annotator models introduced by Davani et al. (2022). Because of its *sep*arated classification *heads*, we refer to this architecture as *SepHeads* in the following to distinguish it from the other approaches modeling multiple annotators.

**Recommender-system inspired models** The second approach, motivated by Gordon et al. (2022), explores *re*commender systems that incorporate LLMs. Such systems rely on two encoder blocks, one for the text (using a LLM) and one for the annotator. This results in two internal vector representations of the input pair of text and annotator ID. To combine these two representations, we use a neural combiner component that performs the final classification. Such recommender style architectures, by encoding annotators by their IDs, can induce representations that generalize across single annotators, thus learning commonalities in the behavior of annotators that have a similar labeling pattern. It is in this sense that the recommender architectures can *relate* perspectives of different annotators.

In the standard recommender approach, the model contains exactly one pre-trained LLM *shared* among all annotators. Hence, we call this architecture *ShareREC*. To explore hybrid approaches emphasizing the more annotator-specific component, we model an option in which each annotator has their own *sep*erate text encoder. This type of model is referred to as *SepREC* hereafter.

We additionally experimented with further architectures that fit in between *ShareREC* and *SepREC*. The methodology and results of these models can be seen in Appendix C.2.

## 4 Experiment Design

### 4.1 Datasets

Due to the fact that work on modeling labeling choices of single annotators is quite recent, so far

there are not many datasets available that have released the data in a way that explicitly contains the labels of individual annotators. We base our experiments on two such datasets that examine different aspects of argument quality.

**CIMT Argument Concreteness Dataset (abbr. Concreteness, Romberg et al., 2022)** The German-language dataset consists of argumentative text units (ATUs) extracted from public participation processes related to traffic planning. These ATUs were categorized into three levels of content-related concreteness: *low*, *intermediate*, and *high*. While ATUs of low concreteness were defined as being vague and lacking specificity, ATUs of high concreteness should provide detailed information.

Each ATU was labeled by five different annotators. Released to account for individual annotation behavior, the authors applied a rigorous annotation process to ensure that discrepancies were due to subjective perceptions and not due to annotation errors.

In our experiments, we use a split of the dataset that was introduced by Romberg (2022). However, in contrast to the original work, we opted not to use repeated $k$-fold cross-validation to minimize the use of computational resources and energy consumption. This decision was based on the reported small deviations in results between different splits.

**Argument Validity and Novelty Prediction Shared Task (abbr. ValNov, Heinisch et al., 2022)** The shared task of the 2022 edition of the Argument Mining Workshop focused on two tasks – predicting once the validity and once the novelty of argumentative conclusions. In the corresponding dataset, consisting of English-language arguments from debatepedia.org, validity captures the extent to which a conclusion is justified given its premise. Novelty captures to what extent a conclusion contains content that is not merely a paraphrase of the premise. In both tasks, annotators had a binary choice but could abstain if they were unsure. A total of five annotators contributed to the annotation process, with each premise-conclusion pair labeled by exactly three annotators.

While the original version of the dataset contains aggregated labels, we disaggregated these labels for the purpose of modeling human label variation. Although not originally designed as a dataset allowing for the study of the labels of single annotators, Heinisch et al. (2022) already emphasized the degree of subjectivity in the annotation of validity and novelty. While about two-thirds of the annotation samples for validity and half of the annotation samples for novelty are non-controversial, i.e., there is complete agreement among the annotators, the labels vary more in the remaining cases, which is a manifestation of their subjectivity. In light of the careful selection of annotators and due to the comprehensive guidelines, we argue it is reasonable to assume that the variations in labels are thus due to individual perspectives.

Given our interest in learning from human label variation, we required samples to be present in the training data for each of the five annotators. As this was not the case for the original split by Heinisch et al. (2022), where two annotators were only introduced in the test split, we re-partitioned the dataset. In doing so, we adhered to the original course of action by avoiding topic overlap between training and the other data, sharing eight of the total 37 topics between development and test data, and introducing seven novel topics in the test data. We also kept the proportions of the original split in terms of the premise-conclusion pairs included. As a result of the non-aggregated approach, the premise-conclusion pairs in the resulting dataset may contain fewer than three labels, as we exclude individual decisions of abstain[2].

Table 1 provides a general overview of the two datasets' distribution amongst training (train), development (dev), and test set, as well as the number of classes.

Going more into detail regarding annotator-individual labels, the proportion within the splits in ValNov - unlike in the Concreteness dataset - depends on the respective annotator. For this reason, Table 2 breaks down how strongly each annotator is represented in ValNov. It shows that the distribution varies greatly, with the predominant annotator covering nearly the entire dataset, while the least represented annotator has assigned labels to only 12% of the examples.

## 4.2 Experimental Setup

We use the transformers-library by Wolf et al. (2020) to implement the text processing units (pre-trained LLMs) in all models. We

---

[2]The annotator-individual split of ValNov is included in the GitHub repository: https://github.com/phhei/RelatePerspectives-sweetspots/tree/main/Datasets/ValNov-new_split

| Dataset | Classes | Train | Dev | Test |
|---|---|---|---|---|
| Concreteness | 3 | 788 | 113 | 226 |
| ValNov | 2 | 746 | 203 | 525 |

Table 1: Overview of the datasets.

| Sub-task | Split | Annotator | | | | |
|---|---|---|---|---|---|---|
| | | #1 | #2 | #3 | #4 | #5 |
| Validity | Train | 739 | 635 | 489 | 204 | 85 |
| | Dev | 203 | 180 | 106 | 75 | 22 |
| | Test | 524 | 443 | 238 | 266 | 68 |
| Novelty | Train | 740 | 642 | 495 | 204 | 87 |
| | Dev | 203 | 180 | 118 | 75 | 19 |
| | Test | 523 | 446 | 240 | 266 | 69 |

Table 2: Annotator distribution among the train/dev/test split for the ValNov dataset.

use the same LLM architecture (RoBERTa, Liu et al., 2019) for all models and settings. RoBERTa has proven to be effective in the prediction of different aspects related to argument quality (Gurcke et al., 2021; Heinisch et al., 2022). We use RoBERTa base variants, namely `roberta-base` for the English ValNov dataset and `roberta-base-wechsel-german` (Minixhofer et al., 2022) for the German Concreteness dataset. Thus, for all models evaluated on the same dataset, the majority of initial weights are equal. During training, we use a class-weighted cross-entropy loss. These class weights are calculated and applied for each annotator separately based on the train split (except for the *Majority*-model where we use the majority label for calculating the class weights). Further choices for model components, in particular for different implementation options for the recommender architectures, were determined during preliminary experiments (Appendix C). For further details on the selection of hyper-parameters, the training, and used computational resources, see Appendix A.

As described in Section 4.1, we use fixed data splits for both datasets (Concreteness and ValNov), having the concreteness task in a setting in which all annotators are equally represented and the two tasks in the ValNov dataset in a setting in which the annotated set of instances differs among the annotators (cf. Table 2). We perform ten runs of training and evaluation for each architecture type, using the same random seeds in the same order (see Appendix A). We report scores based on averages over individual runs.

### 4.3 Evaluation Metrics

How to best evaluate models that do not learn from aggregated labels is an open problem (Basile et al., 2021; Uma et al., 2021b), and various approaches to move beyond majority labels in evaluation exist (Plank, 2022; Leonardelli et al., 2023).

We follow studies on annotator-level models in evaluating against individual annotator's decisions (Davani et al., 2022; Gordon et al., 2022; Orlikowski et al., 2023). However, instead of calculating scores over all individual annotations, we derive scores in two steps: We first calculate the macro-averaged $F_1$ for each annotator separately (i.e., annotator-level scores). As a second step, we take the average of annotator scores as a model's score. This is done for each run of our models, so that the final score is the average of the single-run model scores (i.e., annotator-average scores).

This two-step calculation has several advantages in our setting. By first calculating per-annotator scores, the final score more appropriately reflects how well our models represent each annotator. Additionally, this method allows us to analyze model performance annotator-wise. Based on the range of these scores, we can investigate how much performance diverges between annotators.

## 5 Results & Evaluation

We provide results for the five models considered along the spectrum of architectures as outlined in Figure 1. We evaluate these architectures on two levels using the previously described two-step calculation of model scores. In Section 5.1, we use the annotator-average scores in order to evaluate models across individual predictions (suitable for use cases in which a global indicator of performance is desired). To have a more fine-grained examination (suitable for use cases aiming to maximize argument quality for an individual user) we evaluate against the non-averaged annotator-level scores in Section 5.2. See Appendix B and C for full results and additional recommender configurations.

### 5.1 Annotator-average Results

The annotator-average results are provided in Table 3. All models outperform a naive baseline consisting of always predicting the most frequent label in the training set: "high" for argument concreteness, "valid" for validity prediction, and "not novel" for novelty prediction, which are the same for all annotators. There are major improvements between

+24.7 and +31.69 $F_1$-points in the concreteness task deciding between three classes and some improvements between +7.31 and +29.22 $F_1$-points and between +1.17 and +21.38 $F_1$-points for binary validity and novelty classification, respectively. These diminishing gains over the naive baseline reflect the increasing task difficulty from concreteness to novelty classification (Romberg et al., 2022; Heinisch et al., 2022).

Looking at the different architectures along our architectural continuum, the variation regarding the $F_1$-scores is larger with respect to validity and novelty prediction than with respect to the correctness prediction task.

We find that all three tasks show similar patterns: the models at both ends of our architectural continuum (PerAnnotator and Majority) underperform in general compared to the hybrid models. The PerAnnotator-Models result in 54.57, 44.04, and 41.83 $F_1$-points for predicting concreteness, validity, and novelty on average, respectively. Especially for the latter two tasks where the available training data for some models is reduced to minor amounts (Table 2), which are typically not sufficient to fine-tune a model, this approach falls apart. However, using the Majority model does not increase the performance much in validity and novelty (+3.59 and +1.64, respectively) or even reduce the performance in the concreteness task (−0.75).

Using a hybrid approach increases the performance always with only one exception (SepHeads with 50.83 in the case of concreteness classification). In all other cases, we successfully relate the different perspectives by capturing the common ground of the tasks by also incorporating (and relating) the individual perspectives of the annotators. Having a medium ratio of annotator-specific parameters (Figure 1) yields superior performance. In all three tasks, ShareREC yields the best-averaged $F_1$-scores with 57.83, 65.95, and 62.04 in predicting concreteness, validity, and novelty, respectively, closely followed by the SepREC with annotator-individual text encoders (57.77, 62.08 and 57.03, respectively). The model using separated heads yields $F_1$-scores of 50.83 (concreteness), 53.82 (validity), and 47.04 (novelty).

## 5.2 Minimum and Maximum Annotator-individual Results

Beyond considering only aggregated results in terms of average $F_1$, we also investigate the ex-

tent to which the models considered can represent the perspectives not only of annotators on average, but all of them. For this, we consider the variability in $F_1$ between the user whose perspective (subjective labeling behavior) is captured best and the user whose perspective is captured the worst (i.e., the range). A large range shows that there are large differences in how well we can represent the subjectivity across users.

Table 4 shows the results per model in terms of $F_1$ for the annotators whose behavior can be modelled best (max) and worst (min), respectively. The highest scores across annotators in terms of *min* and *max* are yielded by ShareREC (having $F_1$-scores of between 54.01 and 62.71 for concreteness, 56.12 to 71.16 and 56.80 to 68.85 for novelty), while the other models at both ends of our architectural spectrum (PerAnnotator and Majority) show poor performance, both on the best and worst annotators.

In the concreteness task, where each annotator labeled each sample, we observe that models that include annotator-specific encoders (PerAnnotator and SepREC) can successfully model the subjective text reading behavior of specific annotators. Hence, the PerAnnotator-models have a maximum $F_1$ score of 62.88, which is only outperformed by the separated recommenders yielding the overall best single-annotator $F_1$-score of 67.54, successfully combining the individual view on text for this annotator[3] with the common ground. However, the individual annotator engagement of those models (including the SepHeads model) involves the risk of overfitting individual trends – showing a comparable high standard deviation between the prediction performances ($\approx 6$). Therefore, the best result regarding the minimum annotator $F_1$-score is obtained by the more conservative ShareREC, yielding a score of 54.01.

For validity and novelty, where each annotator labeled a different amount of samples, architectures featuring text encoders that are specific for single annotators fail to reliably predict the behavior of annotators having provided few samples. For example, the PerAnnotator-models have an $F_1$-score of 34.09 and 39.15 for validity and novelty, respectively, taking the most underrepresented annotator into account. This performance is worse than the majority baseline (34.62 validity and 45.24 novelty). Especially for annotators with sparse labels,

---

[3]This annotator has the strongest correlation between label decisions and text length. Only SepREC maximizes the $F_1$-score for this annotator.

|            | Concreteness      | Validity          | Novelty           |
|------------|-------------------|-------------------|-------------------|
| Baseline   | $26.14 \pm 0.00$  | $36.73 \pm 0.00$  | $40.66 \pm 0.00$  |
| PerAnnotator | $54.57 \pm 1.01$ | $44.04 \pm 4.71$ | $41.83 \pm 2.88$  |
| SepREC     | $57.77 \pm 1.18$  | $62.08 \pm 2.25$  | $57.03 \pm 2.09$  |
| ShareREC   | $\mathbf{57.83 \pm 1.44}$ | $\mathbf{65.95 \pm 1.66}$ | $\mathbf{62.04 \pm 0.88}$ |
| SepHeads   | $50.83 \pm 2.31$  | $53.82 \pm 5.79$  | $47.04 \pm 4.22$  |
| Majority   | $53.82 \pm 1.00$  | $47.63 \pm 7.76$  | $43.47 \pm 2.73$  |

Table 3: Annotator-average $F_1$-scores and standard deviation of them for 10 runs.

|            | Concreteness | | Validity | | Novelty | |
|------------|--------------|--------|--------|--------|--------|--------|
|            | Min          | Max    | Min    | Max    | Min    | Max    |
| PerAnnotator | 49.66      | 62.88  | 34.09  | 56.06  | 39.15  | 44.48  |
| SepREC     | 52.14        | **67.54** | 52.59 | 68.91 | 50.37  | 65.73  |
| ShareREC   | **54.01**    | 62.71  | **56.12** | **71.16** | **56.80** | **68.85** |
| SepHeads   | 44.24        | 61.09  | 43.69  | 61.67  | 45.36  | 49.88  |
| Majority   | 47.70        | 58.86  | 36.96  | 54.57  | 38.16  | 47.89  |

Table 4: Highest and lowest annotator-level $F_1$-scores for 10 runs.

it seems thus crucial to share the text encoding components of the architecture. Among the architectures that share components, the recommender systems inspired architectures perform best with respect to predicting the labeling behavior of both under- and overrepresented annotators, yielding the highest min- and max-$F_1$ scores. Recommenders with a shared text encoder perform between $56.12$ and $71.12$ for validity and between $56.80$ and $68.85$ for novelty, yielding the best scores for these two tasks.

When looking at the goal of catching contrastive views and opinions, for example, to detect "hard cases", further insights considering the predicted agreement reveal a weakness of the ShareREC models. ShareREC models stress the common text understanding and learn to predict better-matching majorities that are appropriate for more annotators, resulting in almost no divergent predicted labels per sample. Recommenders with annotator-separate text encoders (SepRECs) model different perspectives much better, having a predicted Fleiss' kappa inter-annotator agreement between $\kappa = 0.73$ (novelty) and $\kappa = 0.79$ (concreteness). However, modeling the individual traits of annotators is challenging, especially for complex tasks such as validity and novelty, explaining the overall worse performance of SepRECs in comparison to ShareRECs in these two tasks.

## 6 Qualitative Analysis: Case Study

The evaluated models show different behaviors that result from how they incorporate individual annotators. These differences can best be illustrated by discussing controversial examples. We consider in particular an example from the Concreteness dataset in which the full range of possible labels is provided by annotators. Table 5 shows such an example of an argumentative text unit about a "bike lane blocked by cars" where two annotators assigned the label "high concreteness", two said it was of "intermediate concreteness", and one labeled the example with "low concreteness". Predictions are taken from models trained with the first random seed (see Section 4.2).

The Majority model, by definition, predicts the same label for each annotator. The predicted value, intermediate concreteness, is plausible in this case at face value: Examining further examples from the dataset shows a tendency for short texts to be annotated as less concrete. However, for two annotators this is an example of high concreteness, despite its brevity. Thus, there is no clear majority label in this controversial case, so this prediction misses three out of five annotators.

ShareREC shows the same prediction pattern as the Majority model. This is in line with the model's discussed tendency to predict uniform labels per example. The model stresses the common ground based on its shared text representation and picks a plausible label.

The PerAnnotator and SepHeads models show more diversity in their predictions. This makes sense, as their architectures contain components for more variation isolated from other annotators via separate classification heads or models. This example underlines, however, that variation does not necessarily lead to more accurate representation of annotators overall. Both models, again, only manage to predict two out of five labels correctly. Thus, they are not more accurate than the uniform predictions.

SepREC, in contrast to all other models, is very close to predicting the actual distribution of labels. For the one annotator it misses, the tendency of the label (less concrete) is correct. SepREC is thus the only model to predict the class "high concreteness" correctly for two annotators. In this example, the model picks up a peculiarity of argument concreteness in public participation related to traffic planning: while generally short argumentative units tend to be less concrete, they might be very concrete to some annotators. For example, annotators might have specific knowledge about local contexts, such as that there are certain bike lanes in a city

| controversial example: *"Bike lane blocked by cars"* | |
|---|---|
| Human Annotations | 0, 2, 1, 1, 2 |
| PerAnnotator | 0, 1, 0, 1, 1 |
| SepREC | 1, 2, 1, 1, 2 |
| ShareREC | 1, 1, 1, 1, 1 |
| SepHeads | 0, 1, 0, 1, 1 |
| Majority | 1, 1, 1, 1, 1 |

Table 5: Example from the Concreteness dataset with each model's predictions. Translated from German: "Zugeparkter Radweg". 0 stands for "low concreteness", 1 for "intermediate concreteness", and 2 for "high concreteness".

that are frequently blocked. This observation is in line with SepREC achieving the highest annotator-individual scores on the Concreteness dataset (cf. Table 4).

## 7 Conclusion & Future Work

In this work, we have proposed a general framework to investigate the performance of different architectures on subjective annotation tasks and similar cases where different legitimate perspectives exist on the appropriate labeling decision across annotators. We have, in particular, highlighted architectures predicting labels for individual annotators along a continuum from fully annotator-specific architectures to architectures that rely only on aggregated annotations that completely disregarded individual annotators. Our main focus has been on hybrid architectures along these extreme ends of the continuum that model commonalities in annotation behavior across individual annotators.

Regarding different aspects of argument quality (concreteness, conclusion validity, and conclusion novelty), we have shown that such hybrid architectures are best suited to classify the range of opinions. Our results show that recommender architectures that use an annotator-shared LLM for encoding the text excel in this setting in terms of maximizing the $F_1$-score averaged over all individual annotators up to $43\%$ over a majority-label model. Nevertheless, further analysis going beyond this averaged score has emphasized the importance of having an annotator-tailored text encoding to capture the different reading nuances in "hard cases", resulting in a variance of predicted classes.

Our work provides a starting point for including human label variation into subjective tasks in the field of argument mining, such as the assessment of argument quality, without denying generally accepted objective concepts of argumentation. Still, further work is needed to fully understand the reasons for disagreement.

## Limitations

We have explored different architectures for modeling human label variation in the subjective assessment of argument quality. Although we can identify clear patterns across languages and tasks, our experimental findings are limited to two datasets that address three different aspects of argument quality. This is due to the fact that only very few argument-mining datasets have been released in a way that explicitly contains the labels of individual annotators. While we believe that we have considered the available sources regarding argument quality, we hope to expand the data basis in the future to generalize findings.

Furthermore, the studied datasets each contain annotations from five individual annotators – allowing us to explore our SepREC models where each annotator requires its own LLM. Settings with higher numbers of annotators will quickly hit boundaries of computational resources, as we need to instantiate a separate pre-trained LLM per annotator which has to be stored on a GPU in order to train the entire model in a suitable time span (cf. limitations of ensemble method in Davani et al., 2022). Future work could explore trade-offs between performance and minimizing the loaded LLM size for each annotator.

Our work assumes that the different labeling decisions of annotators reflect legitimate and valid perspectives. However, it is possible that some disagreements are due to unreliabilities in the annotation process or lack of good guidelines, potentially leading to misunderstandings of the task (Uma et al., 2021b; Paun and Simpson, 2021; Plank, 2022). Issues related to unreliability of annotation processes have been studied exhaustively in the context of crowd-sourcing annotations, and countermeasures have been proposed (see for an overview Paun et al. 2018, 2022c). However, the potential trade-offs between ensuring reliability and preserving diverse annotation behaviors remain less clear. Nevertheless, plausible policies to increase soundness include pilot studies, cautiously selecting annotators and attention checks. Specifically to identify genuine disagreement due to subjectivity, Abercrombie et al. (2023) suggest consid-

ering the intra-annotator agreement, i.e., checking to which extent annotators are consistent with themselves. While the first three policies can only avoid noise a-priori, the latter one can be also used to filter already existing datasets in cases where annotators labeled instances multiple times or multiple similar instances.

Lastly, the train/dev/test splits in our datasets are constructed so that we have annotations from each annotator in each subset. Thus, our results do not apply to settings where a) (some) annotators contained in training are not contained in test, or, vice versa, b) (some) annotators in the test set are not included in training. However, as can been from ValNov results, annotators can be effectively learned from a few annotations using recommender systems. Hence, an application following these findings could ask a new user to annotate a few examples in order to provide predictions that are tailored to this user with minimal effort due to the common ground supporting the decision. Furthermore, the user-tailored predictions can be further refined by asking for more annotations. In addition, models providing multiple predictions per instance can be used for a more fine-grained automatic instance-labeling by providing a range and distribution of labels instead of one single majority label.

## Acknowledgements

This work has in part been funded by DFG within the project ACCEPT, which is part of the priority program "Robust Argumentation Machines" (RATIO), and by the VolkswagenStiftung as part of the "Bots Building Bridges (3B)" project under the "Artificial Intelligence and the Society of the Future" initiative. Julia Romberg is funded by the Federal Ministry of Education and Research of Germany, project CIMT/Partizipationsnutzen of the funding priority Social-Ecological Research (funding no. 01UU1904). Responsibility for the content of this publication lies with the authors.

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

# A  Training Details, Hyperparameters and Computational Resources

In addition to the text-processing units (Section 4.2), also the training loop was implemented using the `transformers`-library by Wolf et al. (2020). For all hyperparameters not explicitly mentioned we used default settings. Maximum sequence length is 512 tokens, with truncation and padding to the maximum length. We train with an initial learning rate of $1e-5$ for all models. Most experiments use a batch size of 8. The only exception is SepREC where we use a batch size of 2 so that the model fits on a single GPU (see below for details on used resources). Each run uses a fixed random seed, most importantly used in weight initialization of the non-pretrained layers: 2923262358, 1842330218, 827634346, 171049425, 991167630, 1070299506, 762227973, 555596930, 1010185121, 419984946

Most experiments ran on a single Nvidia GeForce GTX 1080 Ti (12GB GPU RAM). Only the SepREC experiments ran on an Nvidia A40 (48GB GPU RAM) because of higher RAM requirements due to several LLM instances that need to be loaded simultaneously. Per run, training and evaluation together take on average about 6 minutes for Majority and a single model from PerAnnotator, about 7 minutes for SepHeads, about 10 minutes for ShareREC, and about 50 minutes for SepREC.

As described in Section 4.2, all models use RoBERTa (Liu et al., 2019) to encode text, using `roberta-base` or `roberta-base-wechsel-german` (Minixhofer et al., 2022) initial weights depending on the dataset language.

ShareREC and SepREC are both based on a parallel DeepCrossNetwork (Wang et al., 2021) (see Appendix C for more context) with 3 layers, ReLU activation and 30 appended feed-forwarded features. To encode the user, both ShareREC and SepREC use a feed-forward network with 3 layers (embedding size of 50, ReLU activation) trained with a dropout of 20%.

Based on the development set performance in preliminary experiments, we select a different number of training epochs per setting. The Majority model is trained for 10 epochs. For the PerAnnotator approach, each individual model is also trained for 10 epochs. SepHeads is trained for 7 epochs. SepREC is trained for 12 epochs. We train ShareREC for 20 epochs on concreteness and for 14 epochs on ValNov. Despite these particular choices, we note that the *REC models' development set performance only changes minimally after epoch 12-14.

The majority of parameters in our models belong to the pre-trained language model. Specifically, RoBERTa-base has 125 million parameters. For ShareREC these parameters are multiplied by the number of annotators (five for our setting, so 625 million parameters). We keep the pre-trained model's default output dimensionality of 768. Thus, each classification head (in Majority, PerAnnotator, and SepHeads) adds $768 \cdot 768 + 768 = 590,592$ parameters for a fully-connected layer and $768 * 2 + 2 = 1,538$ (two classes in ValNov) or $768 \cdot 3 + 3 = 2,307$ (three classes in concreteness) for a projection layer. Accordingly, the added parameter counts for ShareREC and SepREC are higher based on 3 fully-connected layers in the user representation's dimensionality (50) and 3 fully-connected layers in the combined dimensionality of user and text.

# B  Full Results for All Individual Annotators

Table 6 contains all annotator-individual results from all tested architectures and configurations.

# C  Recommender Architectures

As already depicted in Section 3.2, our approaches relying on recommender systems contain three core components: one or more blocks processing the text standalone (text encoder), one block for processing the user-id (user encoder) and one block for combining these two encodings and returning the final classification prediction (combiner). For the sake of comparability to our other approaches, we fix the text encoder to a RoBERTa model, having only two degrees of freedom: i) whether there is exactly one text encoder shared among all annotators (*ShareREC*) and/ or a text encoder separately for each annotator (*SepREC*) and ii) whether/ how the separated text encoders are connected to each other. We study these hyperparameter settings in Appendix C.2. For our other two components, we implement several single pieces, which can be seen as additional hyperparameters. To encode the user-ids, we can opt between i) a one-hot encoder or ii) a simple neural feed-forward encoder. As options for combiner, we offer a simple feed-forward neural

*Concreteness*

| | #1 | #2 | #3 | #4 | #5 |
|---|---|---|---|---|---|
| Baseline | 26.06 ± 0.00 | 26.49 ± 0.00 | 25.95 ± 0.00 | 26.06 ± 0.00 | 26.16 ± 0.00 |
| PerAnnotator | 59.98 ± 2.55 | 49.66 ± 2.59 | **62.88 ± 2.10** | 49.70 ± 2.10 | 50.60 ± 1.01 |
| SepREC | **67.54 ± 3.26** | 52.47 ± 1.81 | 60.54 ± 1.93 | 52.14 ± 2.34 | **56.17 ± 3.24** |
| ShareREC | 60.23 ± 2.05 | **54.01 ± 1.77** | 62.71 ± 3.08 | **57.85 ± 1.89** | 54.37 ± 2.39 |
| SepHeads | 61.09 ± 3.30 | 46.09 ± 3.79 | 52.92 ± 6.06 | 49.81 ± 3.59 | 44.24 ± 6.73 |
| Majority | 58.86 ± 1.23 | 47.70 ± 1.75 | 58.61 ± 1.42 | 53.08 ± 1.48 | 50.88 ± 3.25 |

*Validity*

| | #1 | #2 | #3 | #4 | #5 |
|---|---|---|---|---|---|
| Baseline | 37.40 ± 0.00 | 38.30 ± 0.00 | 36.53 ± 0.00 | 36.82 ± 0.00 | 34.62 ± 0.00 |
| PerAnnotator | 47.59 ± 8.21 | 56.06 ± 13.32 | 44.16 ± 10.34 | 38.29 ± 3.75 | 34.09 ± 1.10 |
| SepREC | 66.71 ± 1.33 | 68.91 ± 1.74 | 60.43 ± 1.64 | **61.74 ± 3.87** | 52.59 ± 7.70 |
| ShareREC | **67.93 ± 1.33** | **69.85 ± 0.89** | **71.16 ± 1.61** | 56.12 ± 1.78 | **64.68 ± 5.81** |
| SepHeads | 59.48 ± 10.01 | 61.67 ± 5.42 | 58.78 ± 6.07 | 43.69 ± 5.79 | 45.49 ± 9.29 |
| Majority | 50.47 ± 9.17 | 53.24 ± 10.26 | 54.57 ± 10.54 | 42.91 ± 6.19 | 36.96 ± 5.06 |

*Novelty*

| | #1 | #2 | #3 | #4 | #5 |
|---|---|---|---|---|---|
| Baseline | 40.90 ± 0.00 | 34.89 ± 0.00 | 43.40 ± 0.00 | 38.85 ± 0.00 | 45.24 ± 0.00 |
| PerAnnotator | 44.05 ± 2.54 | 41.26 ± 3.98 | 44.48 ± 2.45 | 40.19 ± 4.36 | 39.15 ± 12.83 |
| SepREC | 55.93 ± 2.65 | **65.73 ± 2.46** | **60.88 ± 1.72** | 52.22 ± 3.22 | 50.37 ± 7.11 |
| ShareREC | **56.80 ± 1.44** | 63.26 ± 1.77 | 59.03 ± 1.18 | **62.26 ± 2.12** | **68.85 ± 2.44** |
| SepHeads | 49.88 ± 6.73 | 48.57 ± 8.40 | 45.90 ± 9.28 | 45.36 ± 5.05 | 45.50 ± 9.63 |
| Majority | 43.35 ± 1.82 | 38.16 ± 2.63 | 45.96 ± 2.19 | 42.02 ± 4.77 | 47.89 ± 4.52 |

Table 6: Annotator-level macro $F_1$ averaged from 10 runs of training and evaluation on the same train/test sets with different seeds. Separate table for concreteness, validity and novelty, respectively.

net processing the concatenation of text-encoding and user-encoding or all variations of DeepCross-Networks as proposed by Wang et al. (2021). We explore various settings with a shared text encoder in Appendix C.1.

## C.1 Hyperparameter Study for *ShareREC*

In order to explore suitable compositions of our implemented single components, we apply a grid search using the following selected anchors after an experimental consolidation phase:

1. For User-Encoding:

   (a) Simple: Feedforward-Neural-Net with 1 layer (embedding size of 25, no activation function) and a dropout of 20%
   (b) Complex: Feedforward-Neural-Net with 3 layers (embedding size of 50, ReLU-activation-function) and a dropout of 20%

2. For Combiner:

   - Simple: Feedforward-Neural-Net with 1 layer (no activation function) and a dropout of 20%
   - Medium: Feedforward-Neural-Net with 3 layers (ReLU-activation-function) and a dropout of 20%
   - Complex: Feedforward-Neural-Net with 5 layers (TanH-activation-function) and a dropout of 20%
   - DeepCross: parallel DeepCrossNetwork with 3 layers (ReLU-activation-function), 30 appended feed-forwarded features

Table 7 presents the results showing that the

| User | Combiner | Concreteness | Validity | Novelty |
|---|---|---|---|---|
| Simple | Simple | $57.41 \pm 1.05$ | $\mathbf{66.28 \pm 0.68}$ | $61.85 \pm 1.35$ |
| Simple | Medium | $55.72 \pm 4.60$ | $56.55 \pm 13.54$ | $49.43 \pm 8.48$ |
| Complex | Simple | $57.30 \pm 1.36$ | $65.48 \pm 1.39$ | $61.69 \pm 1.22$ |
| Complex | Medium | $57.90 \pm 1.32$ | $57.90 \pm 15.27$ | $54.35 \pm 7.84$ |
| Complex | Complex | $\mathbf{58.44 \pm 0.98}$ | $65.99 \pm 1.68$ | $60.70 \pm 1.98$ |
| Complex | DeepCross | $57.83 \pm 1.44$ | $65.95 \pm 1.66$ | $\mathbf{62.04 \pm 0.88}$ |

Table 7: Average and standard deviation of per- annotator macro $F_1$ from 10 runs for different *ShareREC*-hyperparmerters

| $\lambda$ | +shared | Concreteness | Validity | Novelty |
|---|---|---|---|---|
| $-0.5$ | no | $56.93 \pm 0.93$ | $61.26 \pm 2.05$ | $55.34 \pm 3.10$ |
| $0.0$ | no | $\mathbf{57.77 \pm 1.18}$ | $62.08 \pm 2.25$ | $57.03 \pm 2.09$ |
| $0.0$ | yes | $56.30 \pm 1.19$ | $\mathbf{65.67 \pm 0.94}$ | $\mathbf{60.42 \pm 2.38}$ |
| $0.1$ | no | $56.77 \pm 0.97$ | $62.99 \pm 2.08$ | $56.86 \pm 2.41$ |
| $1.0$ | no | $57.01 \pm 0.76$ | $61.83 \pm 2.09$ | $56.77 \pm 1.81$ |
| $2.0$ | no | $57.40 \pm 1.00$ | $61.40 \pm 2.27$ | $56.29 \pm 2.04$ |

Table 8: Average and standard deviation of per- annotator macro $F_1$ from 10 runs for the continuum *SepREC* to *ShareREC*. $\lambda$ is the value mentioned in Equation 1. $\lambda = 0.0$ equals the *SepREC*-setting.

majority of hyperparameter settings yield comparable strong performance (concreteness with an $F_1$-score of $\approx 57$, validity with an $F_1$-score of $\approx 66$ and novelty with an $F_1$-score of $\approx 62$), showing the general success of the recommender architecture and the importance of the text encoder. However, there are some outliers: the combination of a simple user representation and a non-simple combiner do not complement one another well. As observable in Table 7, the simple-user-medium-combiner-setting has the lowest and most unstable scores across tasks, especially for the most complex task of novelty prediction (49.34 points). However, the shallow combination of simple user encoding and simple combiner shows good results and is superior in validity (66.28 points), emphasizing the differences among annotators in the strongest straightforward fashion compared to all other hyperparameter settings in *ShareREC*. However, this combination is outperformed by the more complex setting (complex user encoder and complex combiner) by $+1.03$ points in concreteness (58.44) and by the final selected setting (complex user encoder and DeepCross-Network) by $+0.19$ points in novelty. Despite have only the third best results w.r.t concreteness (0.61 $F_1$-points worse than the best setting), underestimating the conflicting views in that task, this model results in stable and overall superior scores for our tasks in argument quality assessment.

## C.2 Between *SepREC* and *ShareREC*

We additionally explore architectures fitting in between *ShareREC* and *SepREC*. To this end, we introduce the option to loosely connect each text encoder to each other by adding a further loss term as in Equation 1, penalizing the models if the text encoders diverge (too much).

$$\mathcal{L}_+ = \lambda \frac{\sum_{\forall i,j|i<j}(W_i - W_j)^2}{\sum_{\forall i,j|i<j} 1} \qquad (1)$$

Hereby $W_i$ represents the set of all text encoder

weights for the $i$th annotator and $\lambda$ is a hyperparameter regulating the strength of this added loss. While a high positive value of $\lambda$ leads to a tight text encoder connection, only allowing minimal annotator-specific variations, a value next to 0 results in a very loose connection. A negative value would encourage variations between the annotator-specific text encoders. This "smooth" parameter sharing is motivated by Heinisch and Cimiano (2021).

In addition, we explore the private-shared approach by Liu et al. (2017) for our text processing unit.

Following the equation 1, we experiment with following values: $\lambda = [-0.5, 0.1, 1.0, 2.0]$. We note that $\lambda = -0.5$ is even more annotator-specific than *SepREC* since it penalizes similarities between trainable weights of the separated text encoders, emphasizing the differences between annotators. While a value of $\lambda = 0.1$ corresponds approximately to *SepREC*, $\lambda = 2.0$ emphasizes the common ground regarding the text representation and is thus closer to *ShareREC*.

In addition, we experimented with introducing an additional shared text encoder alongside the annotator-specific text encoders (called +*shared*).

Table 8 shows the results using a complex user encoding and the DeepCross-combiner. While the most simple scenario in this continuum (the pure *SepREC*-setting) has the best $F_1$-score of 57.77 in the concreteness task, where all models behave similarly (having minimal $F_1$-scores of 56.93 with $\lambda = -0.5$), modifying the interaction of text encoders has a higher impact in the more text-complex tasks of validity and novelty. While the impact of $\lambda$ is still minor, having a sweet spot at $\lambda = 0.1$ with $F_1$-score between 61.26 ($\lambda = -0.5$) and 62.99 ($\lambda = 0.1$) in validity and $F_1$-scores between 55.34 ($\lambda = -0.5$) and 58.86 ($\lambda = 0.1$), underlining the "common ground"-aspect in text understanding, the combination of a shared text

encoder and an annotator-specific text encoder performs best in validity and novelty prediction (yielding an $F_1$-score of $65.97$ and $60.42$, respectively). However, we observed in the validity task the tendency that this model (including a shared text encoder) relies too much on this shared encoding and propagates more the majority opinion than the other models in the *SepREC*-series. Hence, in terms of capturing different views on validity, we recommend using the model without a shared text encoder and $\lambda = 0.1$, having a better minimal annotator-$F_1$-score of $54.31$ than the counterpart including the shred text encoder ($52.59$).

We finally remark that we observe that the impact of $\lambda$ vanishes primarily at the start of the training process since the core of each text encoder is a large language model which is pre-trained and thus initialized with these pre-trained weights. Hence, future work is to define a dynamic $\lambda$ starting with a high value that is slowly decreasing.