# OpenReview forum: "Architectural Sweet Spots for Modeling Human Label Variation by the Example of Argument Quality: It’s Best to Relate Perspectives!"
_EMNLP/2023/Conference — EMNLP 2023 Main_

### Official Review · Reviewer_kkvu · 2023-08-01

**Soundness:** 4

**Excitement:**

3: Ambivalent: It has merits (e.g., it reports state-of-the-art results, the idea is nice), but there are key weaknesses (e.g., it describes incremental work), and it can significantly benefit from another round of revision. However, I won't object to accepting it if my co-reviewers champion it.

**Missing References:**

The learning from disagreement field is vast. Useful background knowledge should cover agreement statistics, models for annotation aggregation, and lastly, learning from multi-annotated corpora methods. A good survey of this area with many of useful references and resources is the following book:
Paun, Silviu, Ron Artstein, and Massimo Poesio. Statistical methods for annotation analysis. Springer Nature, 2022.

Other useful resources you may want to check:
Aggregating and learning from multiple annotators, EACL 2021 tutorial
SemEval-2023 Task 11: Learning With Disagreements (LeWiDi)
A crowdsourced corpus of multiple judgments and disagreement on anaphoric interpretation, NAACL 2019
Comparing Bayesian models of annotation, TACL 2018

**Paper Topic And Main Contributions:**

The authors experiment with different strategies to train neural models on multi-annotated corpora covering two argument quality classification tasks. The strategies vary from training annotator independent models to 'hybrid' models that share some of their parameters between the modelled data and annotators.

**Questions For The Authors:**

The adopted evaluation strategy computes per annotator metrics which are then averaged to get the model scores.
A. How is this in support of training these models to take decisions? How would you use one of these models in practice? I.e., how would you interpret their output?
B. How is noise being handled by the different methods you considered? (many annotated corpora are quite noisy)

**Reasons To Accept:**

The authors find a bi-encoder model of the text and annotators to consistently get the best results across their evaluation settings. Reasonable baselines were used as well to make the paper a good study.

**Reasons To Reject:**

Limited scope and novelty -- the authors conduct their study on two datasets from the same domain, using existing approaches. Also, to make the 'learning from multi-annotated corpora' strategies practical they should support different sets of annotators with varying amounts of workloads across the train and test data.

**Reproducibility:**

4: Could mostly reproduce the results, but there may be some variation because of sample variance or minor variations in their interpretation of the protocol or method.

**Reviewer Confidence:**

5: Positive that my evaluation is correct. I read the paper very carefully and I am very familiar with related work.

---

> ### Author Rebuttal · Authors · 2023-08-29
>
> Thanks for your review and honoring the quality of our study itself. Also, thank you for the references to improve our related work section.
>
> Regarding the limited scope and novelty: the goal is to explore learning from disagreement in the domain of argument quality, for which it is widely recognized that different people may perceive argument quality differently and yet validly. Despite this generally prevailing consensus, classification models in the field of argument mining have so far (except for the work of Romberg, 2022, which proposes a soft approach to consider a subjectivity score, and thus does not learn directly from the non-aggregated data) been considered from an aggregate perspective. In our study, we rely on the existing non-aggregated datasets for argument quality available at the time of submission.
>
> In our study, we, therefore, decided to give our full attention to this particular area first, since "learning from disagreement" is of paramount importance to the field of argument mining, but has surprisingly received almost no attention. Nevertheless, although the domain is the same, the datasets differ in language (German vs. English) and discussion topics (urban development vs. well-known debate topics such as abortion).
>
> It is true that we don’t invent an entirely new approach here. However, placing these different architectures in such a continuum is novel, consistently evaluated on the annotator level. We think that our results will inspire and help others in their choice of methods regarding the ability to predict labels for specific individuals. In addition, some combinations of modules and ideas are novel, too, as the SepREC (introducing individualistic text encoders) with its connection variants presented in Appendix C.
>
> The review mentions _“different sets of annotators with varying amounts of workloads across the train and test data”_. In order to have a clear study, we rely on datasets with a not too large amount of annotators. It is true that all annotators annotated all samples in the Concreteness-dataset to cover this scenario, but the ValNov-dataset varying amounts of workloads across the train and test data, see Table 2. Here, the total annotated samples as well as the proportions in train/dev/test differ.
>
> ---
>
> # Regarding the questions:
>
> > A. How is this in support of training these models to take decisions?
>
> I’m not sure whether I understand the question correctly. In training, the (class-weighted) cross-entropy loss is calculated over each annotator prediction separately (with the exception of the Majority model). This training is not tailored to the metric but rather general.
>
> > “How would you use one of these models in practice? I.e., how would you interpret their output?”
>
> Our framework has the great advantage of flexible usage. Besides being used in line with the still emerging perspective view of ground truth (see Cabitza et al., AAAI 2023: Toward a Perspectivist Turn in Ground Truthing for Predictive Computing), it can also be applied for conventional single-ground truth machine learning by aggregating the annotator-specific predictions.
>
> Furthermore, our approach can be used for cost-efficient personalized decisions: In our experiments, we could show that the recommender system-inspired architectures were capable of predicting individual preferences with just a few training samples, presumably due to the shared understanding of basic concepts of argumentation among different annotators.
>
> > “B. How is noise being handled by the different methods you considered? (many annotated corpora are quite noisy)”
>
> Disagreement or human variation in labels is often referred to as noise in the conventional sense. In our work, however, this kind of "noise" is the basis on which we work (following Plank, EMNLP 2022: The “Problem” of Human Label Variation: On Ground Truth in Data, Modeling and Evaluation), accepting that there may be several valid labeling choices when it comes to judging the quality of arguments.
>
> However, when noise refers to an unclean annotation process (i.e. due to unreliable annotators), we believe that this is a very universal problem in supervised machine learning, regardless of whether aggregated or non-aggregated data is involved. In our study, we assume that the quality of the data is ensured upfront, as is the case in the datasets we use (see Section 4).

---

### Official Review · Reviewer_XbzC · 2023-08-04

**Typos Grammar Style And Presentation Improvements:** The paper is very well written.
**Soundness:** 4

**Excitement:**

4: Strong: This paper deepens the understanding of some phenomenon or lowers the barriers to an existing research direction.

**Missing References:**

1. Multi-annotator models predate Davani et al - they have been widely  used in work on modelling with subjective tasks such as offensive language detection by, e.g.,  Akhtar and Basile

2. The survey of Uma et al 2021 and the book by Paun et al 2022 discuss a number of highly successive alternative 'hybrid' approaches adding a common backup to learning from crowds models that are not mentioned here, such as

Zhendong Chu, Jing Ma, and Hongning Wang. Learning from crowds by modeling
common confusions. Proceedings of the AAAI Conference on Artificial Intelligence, 35(7):5832–5840, May 2021. URL https://ojs.aaai.org/index.php/AAAI/article/view/16730.

**Paper Topic And Main Contributions:**

This paper explores the impact  of different architectural choices in designing models that can be trained with data containing  subjective variance between annotators, looking in particular at the task of argument quality. The authors argue that such models can be organized along a continuum from models in which labels are aggregated ('annotator agnostic') to models in which every annotator is explicitly modelled ('annotator specific').

The main contributions are

1. the general idea of a continuum from annotator agnostic to annotator specific models

2. the application of these methods to the area of argument quality

3. the finding that 'hybrid' models seem to work best in this setting.

**Questions For The Authors:**

1. Why didn't you consider 'soft label' approaches?

**Reasons To Accept:**

1. Applying techniques for training models while preserving disagreement to argument quality task extends the range of applications in which such models have been tested

2. I like the idea that such models range in a continuum, although see below

3. The key finding - that 'hybrid' models combining annotator-specific with group-derived information achieve the best performance - is plausible and in keeping with the most successful approaches to 'learning from crowds' - see, e.g., chapter 6 of

S Paun, R Artstein, M Poesio (2022). Statistical methods for annotation analysis. Springer Nature.

   (although again, see below)

4. The paper is very well written.

**Reasons To Reject:**

1. similar taxonomies of approaches for learning from disagreeement have been proposed, and more in general comparisons between different approaches on the same have been made, in a number of previous papers, such as Uma et al 2021 that the authors cite:

Alexandra N. Uma, Tommaso Fornaciari, Dirk Hovy, Silviu Paun, Barbara Plank, and Massimo Poesio. 2021b. Learning from disagreement: A survey. Journal of Artificial Intelligence Research, 72:1385– 1021.

   the paper would benefit from some explanation of how the analysis proposed here differs from earlier work other than the different application

2. The proposed model with separate encoders for text and annotators that then alternatively feed into a shared classifier or separate classifiers for each annotator is elegant, but highly reminiscent of a variety of 'learning from crowds' approaches also separately modelling text and annotators, it would be good to have some discussion of the advantages of this particular model.

3. Strangely, the authors do not consider a class of 'hybrid' models which have been shown to be highly effective both in  Uma et al 2021 survey  and  in the recent SEMEVAL Le-Wi-Di shared task (Leonardelli et al, 2023) - namely, the 'soft label' approaches in which the annotator votes are merged but preserving the existence of differences through a probability distribution.

**Reproducibility:**

4: Could mostly reproduce the results, but there may be some variation because of sample variance or minor variations in their interpretation of the protocol or method.

**Reviewer Confidence:**

5: Positive that my evaluation is correct. I read the paper very carefully and I am very familiar with related work.

---

> ### Author Rebuttal · Authors · 2023-08-29
>
> Thanks for your comprehensive review and pointing to some missing related work. We will include these references and discuss in more depth how the architectures investigated by us relate to other architectures in the papers you mention.
>
> Regarding the novelty of and main contribution of our paper, we can confidently claim that so far there has not been a systematic comparison of different approaches along the continuum we propose on the same datasets. This is the main contribution of our paper and we believe that the results showing the superiority of the recommender-based architecture are interesting, and relevant, and novel. In particular, recommender system-inspired architectures have so far not received major consideration in the field of prediction of quality of arguments. Introducing text-separated encoders (or the variant of loose connection as proposed in Appendix C) as a novel explored point on the proposed continuum, too. In this sense, we think that our results will inspire and help others in their choice of methods and future investigations regarding the ability to predict labels for specific individuals, demographic groups, etc.
>
> ---
>
> Regarding your suggestion of considering „soft labelling“ approaches, we think this is a fair suggestion and comment. Certainly, soft-labeling approaches, given that they predict a distribution of labels, could be incorporated into our continuum between approaches predicting a „majority label“ and approaches that incorporate annotator-individuals-specific layers into a model. Conceptually, a „soft label“ approach is comparable to a „majority labelling“ approach if one would use maximum a posteriori inference (MAP), that is choosing the most probable label. Alternatively, one would have to sample proportionally to the distribution. As we evaluate on the level of predictions of labels for single individuals in our framework, both methods are conceptually close in that they make aggregate non-individualized predictions that need to be extrapolated to single individuals. While we thus agree that we could have included „soft labelling“ approaches in the comparison and this could be seen indeed as a weakness of our paper, we do not expect substantially different conclusions and think that the comparison as it stands already is a significant contribution.

---

### Official Review · Reviewer_QBW2 · 2023-08-04

**Soundness:** 3

**Excitement:**

3: Ambivalent: It has merits (e.g., it reports state-of-the-art results, the idea is nice), but there are key weaknesses (e.g., it describes incremental work), and it can significantly benefit from another round of revision. However, I won't object to accepting it if my co-reviewers champion it.

**Missing References:**

-

**Paper Topic And Main Contributions:**

The article explores the challenges of subjective annotation tasks in natural language processing and proposes several approaches to best represent individual and shared perspectives. The findings show that recommender architectures can significantly improve annotator-individual F1-scores. In this regard, the authors create and describe six architectures. Results show that the best architecture used to model human label variation in subjective annotation tasks is called ShareRec, which is a standard recommender approach that combines two representations using a neural combiner component to perform the final classification and contains exactly one pre-trained LLM shared among all annotators.



**Questions For The Authors:**

How can your findings assist researchers intending to begin an annotation task?

**Reasons To Accept:**

The article is very well documented.

**Reasons To Reject:**

The criteria used to determine which architecture is best, however, are only loosely explained: concreteness, validity, and innovation.

**Reproducibility:**

4: Could mostly reproduce the results, but there may be some variation because of sample variance or minor variations in their interpretation of the protocol or method.

**Reviewer Confidence:**

3: Pretty sure, but there's a chance I missed something. Although I have a good feel for this area in general, I did not carefully check the paper's details, e.g., the math, experimental design, or novelty.

**Typos Grammar Style And Presentation Improvements:**

-

---

> ### Author Rebuttal · Authors · 2023-08-29
>
> Thank you for highlighting that our paper is clearly and well-written.  Regarding your concerns on which criteria we have used to select the best architecture, we would like to emphasize that we have opted for a standard evaluation / benchmarking setting, comparing the different architectures on established datasets in the field of argumentation (see prior work by Romberg et al. (2022) and Heinisch et al (2022) as stated in Section 4). We have opted for tasks in the area of computational argumentation and in particular focus on dimensions related to the quality of arguments. The prediction of quality of arguments has been shown to be a challenging task which is subjective, leading to substantial disagreements between annotators. The particular choice of task / datasets regarding the prediction of concreteness, validity and novelty is due to the fact that these tasks feature non-aggregated datasets, that is datasets where the individual annotations of different annotators are explicit. This is not the case for many datasets and explains why for our study we selected those.
>
> ---
>
> Regarding your question, _“How can your findings assist researchers intending to begin an annotation task?”_, our findings exemplify the value of using non-aggregated labels in model development for subjective tasks. Hence, to assist researchers intending to begin an annotation task that bears subjective traits we recommend NOT merging all annotation labels into one per sample but providing single annotation decisions for each sample.

---

### Meta-Review · Area_Chair_tQCg · 2023-09-20

**Recommendation:** 4

**Metareview:**

This paper explores the impact of different architectural choices in designing models that can be trained with data containing subjective variance between annotators, looking in particular at the task of argument quality. The authors argue that such models can be organized along a continuum from models in which labels are aggregated ('annotator agnostic') to models in which every annotator is explicitly modeled ('annotator specific').

The shared perspective of the reviewers wrt this paper include:
* the paper is well-written and the approach is sound and explored with reasonable baselines
* the contribution of a continuum of annotator agnostic and annotator specific models and the findings on which models yield best results are interesting and valuable
* the novelty of the proposed approach is not clear

As the approach is accepted to be quite sound and there are no scientific or technical issues raised in the reviews, I tend to base my score strongly on this. This is encouraged also by the interesting results provided and the potential application to other tasks. The issues of novelty and comparative discussions wrt existing approaches seems one that can be resolved through elaboration in a camera-ready (as done so in the author response).

---

### Decision · Program_Chairs · 2023-10-07

**Decision:**

Accept-Main

**Comment:**

This paper explores the impact of different architectural choices in designing models that can be trained with data containing subjective variance between annotators, looking in particular at the task of argument quality. The authors argue that such models can be organized along a continuum from models in which labels are aggregated ('annotator agnostic') to models in which every annotator is explicitly modeled ('annotator specific').

The shared perspective of the reviewers wrt this paper include:
* the paper is well-written and the approach is sound and explored with reasonable baselines
* the contribution of a continuum of annotator agnostic and annotator specific models and the findings on which models yield best results are interesting and valuable
* the novelty of the proposed approach is not clear

As the approach is accepted to be quite sound and there are no scientific or technical issues raised in the reviews, I tend to base my score strongly on this. This is encouraged also by the interesting results provided and the potential application to other tasks. The issues of novelty and comparative discussions wrt existing approaches seems one that can be resolved through elaboration in a camera-ready (as done so in the author response).